# Impact of Strategy Change on Business Process Management

Peter Bubenik *, Juraj Capek, Miroslav Rakyta ⬤, Vladimira Binasova ⬤ and Katarina Staffenova ⬤

Department of Industrial Engineering, Faculty of Mechanical Engineering, University of Zilina, 010 26 Žilina, Slovakia
* Correspondence: peter.bubenik@fstroj.uniza.sk; Tel.: +421-041-513-2719

**Abstract:** In the pursuit of economic survival in the current competitive conditions with the aim of long-term prosperity and sustainability in the market, many companies today approach significant strategic changes in the management of their business. The purpose of this study is the design of a systematic procedure for implementing strategy changes into internal business processes for a project-oriented production type of organization. The proposed methodology contains steps where the selection and verification of key performance indicators at individual levels of management takes place. Furthermore, their monitoring and quantification of the impact of the change in strategy on internal company processes. The result of the study explains how the management can monitor and evaluate the chosen processes in accordance with the fulfilment of the chosen strategy of the company, which supports the systematic introduction of changes in the processes with the aim of sustaining the company's performance.

**Keywords:** advanced industrial engineering; strategy; management; business processes; business performance; key performance indicators

## 1. Introduction

Due to the constant growth of input costs (personnel costs, energy, input materials, etc.), many Slovak companies have found themselves in this difficult position, having to wage tough competitive battles both on domestic and global markets. Competitive advantage can be created by free disposable capital, a traditional position on the market, or long-term management experience; above all, you need the ability to use radical innovation of internal processes, tools for managing these processes, and online monitoring of selected parameters as a very effective weapon.

The key to success is not the micro-management of individual operations, but focusing on the main internal processes in the company in such a way that improvements can be achieved in terms of critical weight of performance; these include costs, quality, services, deadline fulfilment, and above all, orientation to the customer and their problems and the challenges it encounters in the implementation of its business models.

The company's strategy guides all management decisions from the front line. Strategic plans act as a road map that helps businesses achieve the big vision of their owners and top managers in practical ways. A change in an organization's strategy can change the way an organization operates, changing everything from the organizational structure to the daily routines of employees. When changing the strategy, it is vitally important to realize the individual correlations between the sub-strategies themselves and the internal processes. Not all effects of change are positive. Internal employee resistance can be a ma-jor obstacle to effective change implementation, as some people strongly resist any kind of change to the status quo or daily routine. There is also always the possibility of failure in new initiatives, leaving the company in a worse position than it was before the change.

Fundamental factors that fatally affect the success of implementing strategy changes into internal company processes have been identified. A survey of 100 manufacturing companies in the Slovak Republic revealed the following factors (Figure 1).

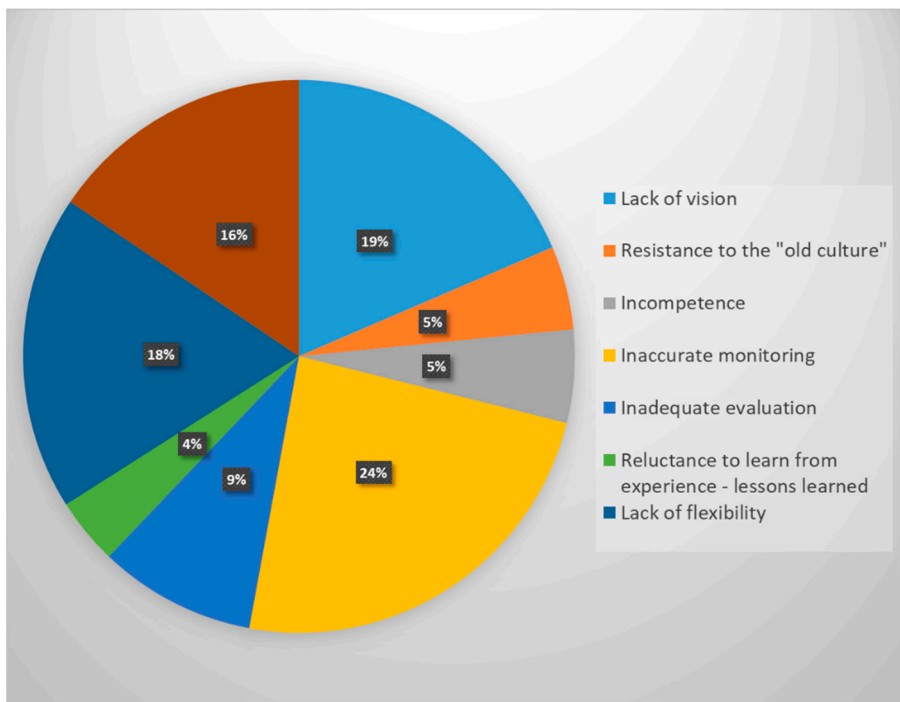

**Figure 1.** The fundamental factors that fatally affect the success of implementing strategy changes into internal company processes.

A project-oriented production organization that pursues a single goal, namely the successful implementation of individual projects in terms of costs, meeting deadlines, covering overheads, and of course, expected profitability. With the successful implementation of individual projects and the non-exceeding of individual budgets in the administration departments, the continuous fulfilment of the relevant business plan of the company in the relevant year is also guaranteed. The preparation process of internal process changes in view of the company's changing strategy consists of several parts that are essentially independent, but mutually connected (see Figure 2).

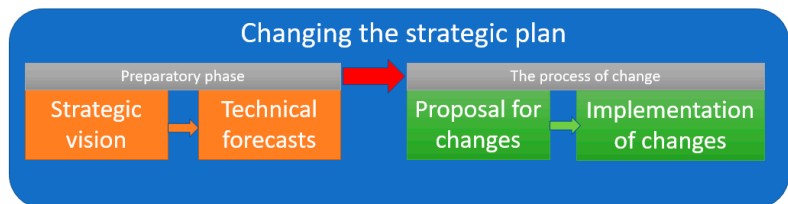

**Figure 2.** The relationship between strategic plan changes and process change.

A change in a strategic plan is a change that is based on the results of strategic planning, and forecasting is a result of technical forecasts. In this process, the healthy core of the company must be used as a basis for achieving set goals, supporting strategy creation, and defining priority changes in internal processes. The process of changing internal processes due to changes in the company's strategy is divided into two phases: design of changes (concept, plan); and implementation of changes. The proposal is the beginning of the change process and contains the position of the project in the company, value indicators, and also all managerial, operational, social and technological changes. Implementation is a process in which we develop plans for internal process changes that are gradually implemented, and at the same time they are monitored, evaluated and corrected with appropriate measures.

The proposed methodology represents the timely identification of changes in monitored key indicators through monitoring, which will enable timely intervention in processes and the elimination of negative impacts so that currently ongoing projects are not threatened.

## 2. Background

Digital transformation reveals new ways for an organization to stay in touch with customers and consumers and thus create value for them [1]. Most consumers behave subconsciously or unconsciously [2]. Marketers try to understand how individual features of a product contribute to the overall evaluation [3].

Today, most authors and managers consider strategic management to be a complex, internally *structured*, and continuous process [4,5]. A simple model of strategic management was presented in his work by [6] considering the continuity of the basic phases. It is possible to state that strategic management is an iterative process, gradually passing through individual phases and their steps; a process that is constantly and continuously ongoing. Automotive and component manufacturing companies form a tandem of recognized prestige in terms of competitiveness and results [7].

Strategic management integrates multiple procedures and analyses which evaluate the organization's results and adapts them to changes in the external environment. It is a relatively complex process that involves several managers in the organization and affects the actions of all employees [6]. The rapid progression of globalisation has driven companies to search for new approaches to improve their performance by producing products at a much lower cost, timelier and with superior quality [8].

The identification of research gaps from systematic reviews is essential to the practice. It is important to address that business process management will continue to be a crucial tool for any organization that is trying to improve its operations. New technologies will emerge providing new opportunities to obtain automational effects, informational effects, and transformational effects. The key question is: "How will digitisation change jobs and work profiles?" The authors of [9] claim their analysis as follows: The extent of computerisation in the twenty-first century will thus partly depend on innovative approaches to task restructuring. Restructuring tasks, and more broadly operations, are exactly what BPM is concerned with [10] in his study, concluding that business process management is developing a stronger strategical perspective which has the potential to support digital strategies of customer engagement and digitized solutions.

Figure 3 below describes the steps of the strategic management process as a continuous process.

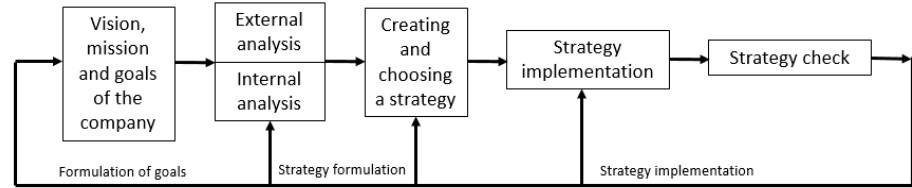

**Figure 3.** Steps of strategic management.

**The first step of strategic management** is the formulation of the strategic intent. It primarily contains the determination of the business vision and mission of the organization, and based on them the long-term strategic goals of the organization are established. Briefly, the vision of the company can be characterized as a time-bound description of the form into which the company wants to transform at the end of the planned period. The vision represents an attractive picture of the company in the distant future. The mission is then a time-bound formulation of the company's future focus, while the mission primarily answers the questions "Who are we?", "Why do we exist?", or "What is the basic meaning of our existence?". The answers to these questions create the basic values of the organization. The strategic goals of the organization should be based on the already established vision

and mission of the company. Strategic goals define the end state towards which the organization's vision is directed. Their purpose is to define a measurable outcome of the vision. Part of this step is the analytical phase, i.e., a detailed analysis of the internal and external environment of the organization [11].

The analysis of the internal environment of the organization aims to identify the strengths and weaknesses of the organization in relation to the established strategic goals, vision and mission of the organization. The purpose is also to identify key competencies and key vulnerabilities of the organization [12–14].

The analysis of the external environment focuses on identifying those opportunities and threats of the external environment that could affect (positively or negatively) the achievement of the set strategic goals. There is a relatively wide range of tools for analyzing the internal as well as the external environment [15].

The last part of the formulation of the strategic intention is the so-called strategic synthesis, for which a SWOT analysis is normally used and serves to determine the strategic position of the organization.

**The second step of strategic management** is the creation of the strategic plan itself. Based on the performed SWOT analysis [16–20] variants of the strategy are determined, and subsequently the selection of a suitable variant of the strategy is based on the synthesis of information from the internal environment of the organization as well as from the analysis of the external environment. After choosing a suitable variant of the strategy, the results are balanced with the original ideas about the vision of the company, its mission or established strategic goals. If the results of the analysis proved that the originally intended vision and strategic goals are unrealistic (oversized or undersized), it is necessary to adapt the vision and strategic goals to these results. Subsequently, it is necessary to specify the strategy for different levels of the organization. For all strategies, it is expedient to create an implementation plan according to the chosen management method, such as BSC, and EFQM. The result of this step is a detailed strategic plan for the organization [21].

The Balanced Scorecard (BSC), a strategic performance-management tool that brought strategy and clarified major organizational objectives for companies' agendas [22], significantly contributed to this process. Many leading companies began to adopt the BSC when they found that it allowed them to improve performance, linking all organization members in a joint effort to achieve the organization's overall goals and objectives [6–9]. The ability to achieve ambitious goals depends, as such, on organizational capabilities for learning and growth [23–27].

**The third step of strategic management** is the implementation of the strategic plan in the day-to-day activities of the organization. In this step, the strategic plan is elaborated into tactical and operational plans which affect the day-to-day activities of the organization and all management functions are subordinate to them—planning, organizing, leading and controlling.

**The fourth step of strategic management** is strategic control, which enters the process of strategic management. Each of its steps allows measuring the ongoing results of the entire process and correcting individual steps. At the same time, it brings continuity to the entire process and makes strategic planning a never-ending process.

*The Management of Change*

The driving force behind changing a company's strategy is always the desire to move the company to a higher level or improve its economic results. An important factor is to realize that a change in internal processes caused by a change in strategy is always a relatively large intervention in the internal functioning of any company. Therefore, it is necessary to focus on a correct and meaningful awareness of what changes need to be made and how to understand these changes in the wider context of all processes in the company.

Categorization of changes in internal processes caused by a change in the company's strategy is a process in which changes can be divided into three basic groups (see Figure 4) according to the severity of the influence of depth on process redesign or adjusting the strategy.

| Adaptive changes | | Transformational changes |
|---|---|---|
| Process optimization | Process redesign | Strategy redesign |
| - Optimization of individual processes<br>- Decentralization within existing processes and organization of building optimized processes | - Company-wide organizational implementation of the process<br>- Issues beyond the scope of processes remain unprocessed | - Company-wide reorientation of procedures and construction<br>- Integration of process-extending and external influencing factors<br>- Orientation to a new customer<br>- New product<br>- New market strategy<br>- New setting of internal culture<br>- Team work |
| - Isolated monitoring of processes without monitoring key competencies and priorities<br>- The changing framework conditions are not taken into account | - Changing external framework conditions in the company, with customers<br>- There is a lack of implementation of activities aimed at the overall change of the company's orientation | - Basal review of all used resources related to complex business activity<br>- A total/radical change in the orientation of the company's business towards external conditions |
| Limited/Tactical | High | Strategic |
| Assumed permanence of changes | | |

(Signs — Evaluation row labels on left side)

**Figure 4.** Essential distribution of changes according to the severity of the impact of process changes.

Change management is influenced by resource settings and the selection of key performance indicators (KPIs) [28]. Companies process volumes of data, looking for suitable indicators so that they can make decisions and manage the company's internal processes. It is a process of guiding organizational change to implementation, from the earliest stages of conception and preparation, through implementation and finally to resolution. Change processes have a set of initial conditions and a functional endpoint. The process in between is dynamic and takes place in stages.

From the point of view of company performance companies monitor financial performance indicators, and the most common indicators that are implemented in IS or are provided within management information systems (MIS) are:

- Evaluation of the management result through the income statement, or balance sheets;
- Assessment of financial flows [29–32];
- Indicators resulting from the company's financial statements;
- Activity indicators: inventory turnover time, debt collection time, liability maturity date, total asset turnover;
- Liquidity indicators: immediate liquidity, current liquidity, total liquidity;
- Indebtedness indicators: total indebtedness, self-financing coefficient, credit indebtedness;
- Profitability indicators: profitability of total assets (ROA), the profitability of equity (ROE), the profitability of sales, the profitability of costs, return on investment (ROI);
- Market value indicators: profit per share, dividend yield;
- Cost indicators: total cost, personal cost.

To the most used indicators from the point of view of the owners, e.g., belong to:

- Economic value added—EVA (Economic Value Added);
- Value added by the market—MVA (Market Value Added);
- Added value for shareholders—SVA (Shareholder Value Added);
- Profitability of net assets—RONA (Return of Net Assets) and others.

Based on the study [33] the challenge for any company is to identify the most relevant model that can be adapted to the contexts of the business.

## 3. The Impact of Strategy Change on Business Process Management

Modern business organizations face dynamic changes in their management environments, and many firms actively consider adopting information technology (IT) to adapt to these changes [34]. Every company management in this hectic time faces the challenge of maintaining stability and achieving a certain level of predictability of the company's performance [35–39]. To meet new challenges and business interests organizations must constantly monitor, evaluate and adjust their strategic initiatives. When a new strategy

needs to be implemented [40], it is usually up to managers to ensure its successful implementation. Information on how to successfully implement strategic change into processes usually contains several step-by-step procedures that are very general.

The subject of the investigation was the realization of how the change affects the strategy and internal processes of the organization. This is done by describing what change is, discussing categories of change, external drivers of change, and perceptions of change initiatives as negative or positive [30,31]. The consequences of changes in strategy can take many forms: for example, it brings different challenges to different people depending on their position in the organizational hierarchy.

Whenever we change our strategy, we encounter the following questions:

- What is changing?
- Why is it changing?
- How will it affect our area?
- How will it directly affect us?

Each of these questions basically has a common denominator and that is internal processes, and all of them will be transformed into the change in these processes. It is therefore important to examine the impact of a change in strategy on the change in internal processes, so that it is possible to better predict the development after the change in strategy on this basis.

Research on the impact of strategy changes has shown that strategy change is a process that relates to the overarching goals and objectives of a business. Strategic decisions influence which business area the company operates in and for whom it benefits. It is also important how the company functions internally and what factors influence changes in the internal process (Table 1). It is very difficult to predict exactly what will happen when an organization changes its strategy, but companies experience several common positive and negative effects when they undergo a strategic transformation.

**Table 1.** Factors affecting/not affecting the success of internal process change.

| Factors Affecting the Success of Internal Process Change: | Factors Not Affecting the Success of Internal Process Change: |
|---|---|
| <ul><li>The dynamics of growth and expansion of the company are very closely connected with the strategy of the company.</li><li>Analysis of strategic value-generating processes.</li><li>Realistic assessment of the current situation, not concealing reality.</li><li>Determining strategic goals that correlate with the monitored process.</li><li>Clear definition of the business plan.</li><li>Consideration of business activities.</li><li>Customer involvement (customer perspective on process change).</li><li>Previous process reorganizations.</li><li>Motivated team. Experiences.</li><li>Execution of these activities by a dedicated key team on a full-time basis and not as part-time activities in addition to daily operations.</li><li>A detailed timeline with clearly defined milestones that can be evaluated, including a process validation pilot test.</li></ul> | <ul><li>Unrealistic setting of goals and expectations.</li><li>Missing or insufficiently detailed "how it to do" methodologies.</li><li>Weak involvement of top management.</li><li>Unpreparedness and inflexibility for changes.</li><li>Excessive concentration only on cost reduction.</li><li>Efforts to make changes only in individual departments without the possibility of cross-implementation through several departments or the whole enterprise.</li><li>Too many parallel implementations of changes in internal processes started.</li><li>Incorrect graduation and enforcement of changes among employees.</li><li>Insufficient connection to the latest IT technologies, weak IT support, undefined IT strategy.</li></ul> |

Based on the research, we can assume that the problem of reaction to the external environment does not lie in external factors but is rather related to how the management of

the company can process these factors, especially if the external environment is constantly changing and dynamic.

Each input for the needs of strategy change is generally based on three aspects of Figure 5, which form a strong stimulus for changing the corporate strategy in the context of changing internal company processes.

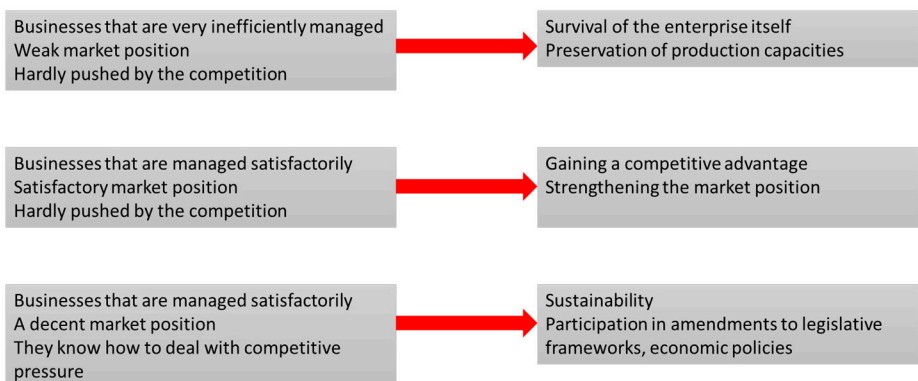

**Figure 5.** The status of the enterprise and the target state of the change.

## 4. Methods

Business Process Management is essential for any organization that operates based on the division of labour, handoffs of creatives, context switches, responsibility and knowledge gaps, and other sources of inefficiencies and defects. Various research methods have been used to research the topic:

Questionnaire—serves to find out the influencing factors on the success/failure of implementing strategic changes in internal company processes by manufacturing companies in Slovakia.

Interview—at MATADOR Automation s.r.o. in Dubnica nad Váhom, a structured interview was conducted with the director of the company.

Analysis and synthesis—the method was used in the summarization of theoretical knowledge, as well as in the design part of the work.

Abstraction and concretization—the method was used in the design of the methodology for the implementation of strategic changes into business processes.

Induction and deduction—the method was used in the design of implementation of strategic changes in business processes.

Comparison—the method was used in the design part when designing the analysis for assessing the cost of projects.

Case study—the method was used to verify the design of the methodology.

Observation—the method was used in cooperation with a production company.

## 5. Proposal of Steps for Implementing Strategy Changes into Processes

The process of implementing a change in strategy is an essential question of how to implement this—sometimes very complex—process for every business without disrupting the smooth running of the business so that only necessary internal processes are changed. Based on the studied sources and the experience of managers from practice, the methodological procedure of implementing the change is divided into the following steps and the relevant parts of Figure 6.

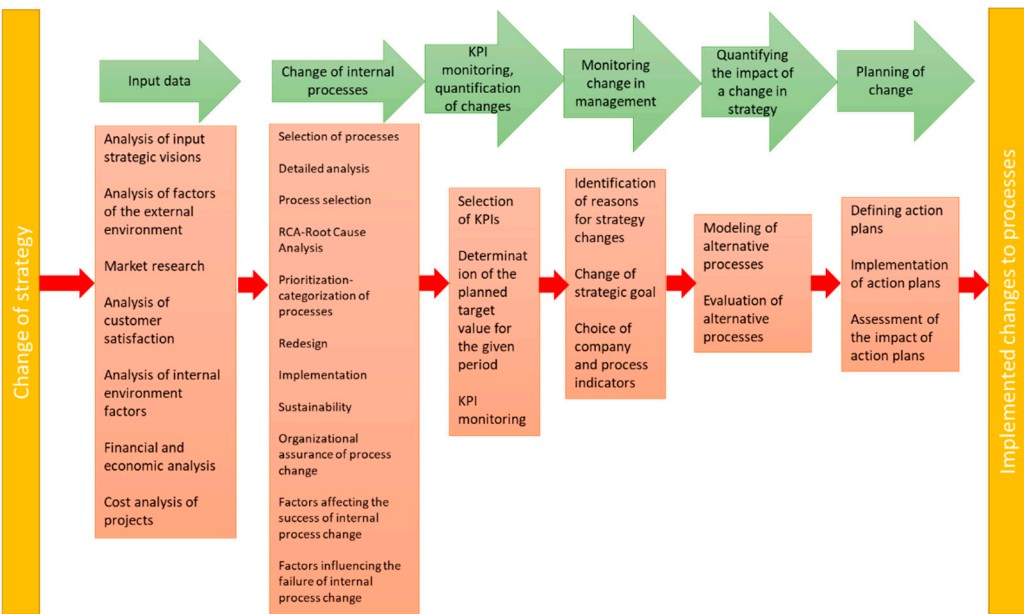

**Figure 6.** Proposal of steps for implementing strategy changes into processes.

*5.1. Input Data*

The goal of the first step is to define the input data necessary for the implementation of individual analyses and steps of the methodology. The methodology begins with a strategic intention, i.e., by defining the organization's goals and using them as a benchmark to measure performance and progress. An organization's vision and direction should be specific, actionable and measurable, not broad. This is the point where companies outline their future business focus, be it profitability, shareholder wealth or market leadership. Basic analyses at the level of the proposed methodology include:

**The analysis of the input strategic visions** is proposed as a process of quick forecasts—brainstorming at the highest level of the company's management and can concern different parts of the company. The result of this process is the creation of strategic visions, which will provide us with a quick overview of business conditions, estimate the company's strengths and weaknesses, and reveal opportunities for change. The basic goal of strategic visions is to set the goals of the company, to create an idea of the future in the relevant field, to establish structures for the realization of visions and to generate sufficient support for the implementation of changes in internal processes. Strategic visions enable company management to define and communicate the cause-and-effect relationships that provide the value proposition of their business, and the results report is an effective tool for implementing and monitoring the company's strategy. Therefore, a system based on strategic visions provides some guidance for gathering and communicating information about value creation.

**The analysis of the factors of the external environment** and the position of the company on the market is for the purpose of a detailed examination of the external environment, where it is possible to implement the following analytical tools:

- Market analysis;
- Porter's 5F;
- SWOT analysis.

The design of the tools is based on empirical experience, when during the implementation of several analytical tools the best results were achieved precisely with the help of the mentioned analytical tools, due to the easy education of the entire team and with regard to a very complex view of the company and its position in the competitive environment.

**Customer satisfaction analysis** collects data through a survey or questionnaire to identify the behaviours that lead to satisfied or dissatisfied customers. The level of customer

sustainability and its impact on profits can be expressed in graphs for better visualization. In this data, we can detect trends in dissatisfaction and correlations with data on customer loyalty behavior. Findings from a customer satisfaction survey help the business develop action plans to address weaknesses and identify people responsible for improving processes. Regular and repeated evaluation of the process usually keeps it on track.

**The analysis of the factors of the internal environment** is an important process that allows the management of the company to estimate the willingness and ability of the company to initiate changes in internal processes within the framework of the strategy change. This helps predict the time required to implement these changes. The primary goal of technical forecasts is to prepare the company for fundamental changes:

- Defining the scope of required changes;
- Full understanding of the direction and impacts of changes;
- Preparing the company for technical changes such as changes in internal processes, information system planning, defining new internal processes.

To successfully understand the overall impact of the strategy on the change in internal processes, a survey carried out in 100 companies states that it is necessary to consider the following facts in the process of technical forecasts:

- Business plans of the company;
- Key account management;
- Perception of the company by customers;
- Carrying out measurements—monitoring in the company;
- Organizational structure and its adaptation options;
- Company culture;
- Operational possibilities of the company;
- Capacity options of the company;
- Infrastructure and use of internal technological possibilities.

**The financial and economic analysis of the company** will provide an assessment of economic trends, the setting of financial policy, the compilation of business activity plans and the identification of projects. This is done through the synthesis of financial numbers and data. During the research, the following ratio financial indicators were verified to enable comparison with other companies' data or with the company's own historical performance: REVENUES—income, EBITDA—profit before interest, taxation, depreciation and amortization PBT—profit before tax, NET DEBT—net debt, ROE—return on equity.

These ratio financial indicators are comprehensive and clearly interpret the company's performance and serve for a quick and accessible comparison of data obtained from publicly available sources within the framework of a competitive benchmark on the EU market and within the framework of the application of the EU uniform directive on financial indicators of companies.

**Cost–benefit analysis** is the process of comparing the forecasted costs and benefits (or opportunities) associated with a project decision to determine whether it makes business sense. The cost–benefit analysis involves adding up all project costs and subtracting that amount from the total projected benefits of the project or decision. If the expected benefits outweigh the costs, it can be argued that the decision is a good one. If, on the other hand, the costs outweigh the benefits, then the company may want to reconsider its decision or project. Cost–benefit analysis is a form of accurate data-driven decision-making.

Based on the research of the implemented projects, a cost tracking group was proposed and divided into the following categories:

- Material complexity of projects (MN);
- External cooperation (KOOP EXT);
- Internal performances (INT);
- Other direct costs such as transportation, travel, rent, etc. (OPN);
- Correct overhead (SR);
- Ratio of internal/external performances (INT/EXT).

The proposed categorization of costs is based on a comprehensive assessment of project management and controlling monitoring of incoming costs in specific parts and phases of the project.

*5.2. Methodology for Changing Internal Processes*

The goal of the second step is to change internal processes resulting from a change in strategy. The change process is designed in individual subsequent steps in Figure 7 and will be described as follows.

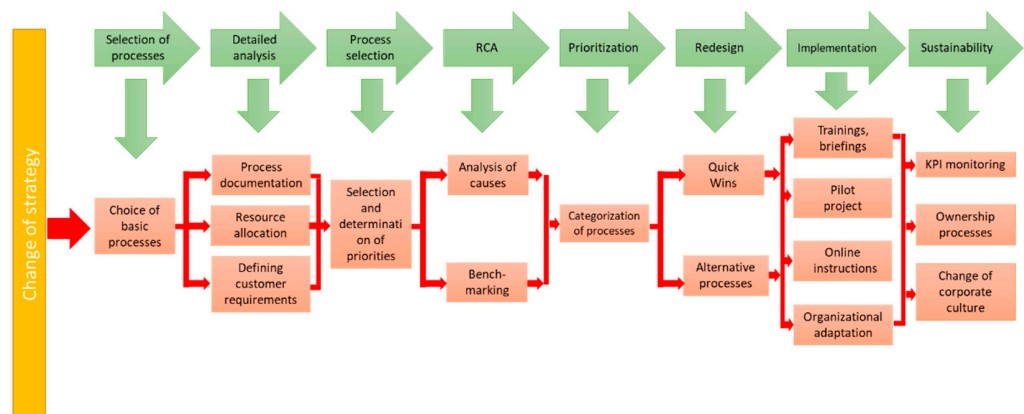

**Figure 7.** Graphic representation of the proposed procedure for changing internal processes.

**Process selection** is carried out throughout the company with the participation of all departments. In this process, it is very important to consider all internal aspects, so that no relevant process is left behind, otherwise there is a risk of failure of the entire main process. If all relevant processes do not undergo changes, the resulting main process will never function comprehensively as a whole. One of the more formal tools that can capture processes in a company is the SIPOC process map.

**A detailed analysis of business processes** helps to increase the efficiency of the process. It evaluates how well the process achieves its ultimate goal. Business process analysis identifies and examines each part of the structure, including the process itself, stakeholders, information exchange, and others. Accordingly, it can help to identify potential improvements within the process, making it easier to execute a reengineering initiative sometime down the line. Currently, software systems are available for design support, analysis and process modeling such as QPR ProcessGuide (QPR Software), ProVision WorkBench (Proforma Corporation), FirstStep Designer (Interfacing Technologies, LBMS), ARIS—Toolset and ADONIS.

The analysis will result in a limited group of processes, which will be properly documented. These processes will have assigned resources and will have defined requirements from customers. In the analysis, it is necessary to focus on three main topics:

- Change in the way the process is carried out due to the change in strategy;
- Determining whether the process is operating at maximum potential capacity;
- Determining whether the process should be improved or redesigned.

**The selection of the priority process** will be carried out on the basis of the priorities that result from a detailed analysis of the processes, where all the factors mentioned above are taken into account. The point of process selection is to understand how you can improve your internal process. Using the findings from the previous steps, deficiencies can be addressed, and potential improvements can be made. The kind of enhancements that would work for a particular business always depends on the particular situation, and when choosing a process the long-term effects of any changes must not be forgotten.

**RCA—Root Cause Analysis** has the task of systematically identifying the root causes of the problem and an effective way for an adequate response to the given cause. The

analysis is based on the dogma that it is best to find a way to prevent problems. During the RCA analysis, the factors affecting the specific process that is the subject of the solution are identified. The goal of this analysis is to reveal the underlying causes of deviations and errors so that we reduce their occurrence in the future or their impact.

**Prioritization** will determine the time schedule for deploying new or optimized processes to ensure the successful implementation of changes in the company's internal processes. An important factor in setting priorities is the awareness of the financial complexity of each such implementation. A large part of the company's budget will probably be spent on material and personnel support for the running of the company, so it is necessary to check whether there is enough funding for each of these activities. If there is not, some priorities probably need to be adjusted.

**The redesign** is the rapid implementation of immediately feasible improvements, the so-called Quick-wins processes and the creation of alternative processes. However, with these changes, one must consider the factor that although something may seem wonderful in the short term, it can be disastrous in the long term. For example, increasing the speed of a process can double the rate of defect risk, which brings the company back to square one in terms of process improvement and reduces the company's profitability as the cost of poor quality increases.

**Depending on the findings**, a decision needs to be made whether small, modest changes are made to the process or whether it needs to be completely overhauled. While "re-digging the process" later may require more resources and time, the effects of elapsed time can be very different.

**The implementation phase** occurs when the proposed solutions are transformed into actionable processes that can be verified through process simulations, prototypes and pilot projects. A very important part of the implementation process is the education phase, which must be included in the implementation phase in order to take advantage of the parallel synergistic effect. This sub-phase of education should be supported by already processed online instructions and procedures, which of course can still change in this phase. A critical part of the implementation process is a detailed timetable, with marked partial milestones so that all necessary stages have their time and resources allocated.

**Sustainability** is the last key phase of changing internal processes, and the sustainability of these changes which we can ensure through constant KPI monitoring. Through constant monitoring and measurement, the process of improvement and improvement of the internal company process is continuously initiated. "Ownership" of processes is also an important element of successful sustainability, where it is clearly defined for all involved who is responsible for which process, and who is responsible for updating the given process so that this process always follows the latest trends and needs of the company.

**Organizational assurance of process change** is the basis of successful process change and its sustainable deployment across the entire company. It is desirable to build teams that will be precisely optimized for the solution of a specific process, to take full advantage of the synergistic effect that can be found in several processes (Figure 8). Ideally, build interdisciplinary work teams so that the team includes all those who in any way touch the process to be optimized. In such teams, the so-called "professional blindness" is often a very determining element when introducing any changes in processes that are already running.

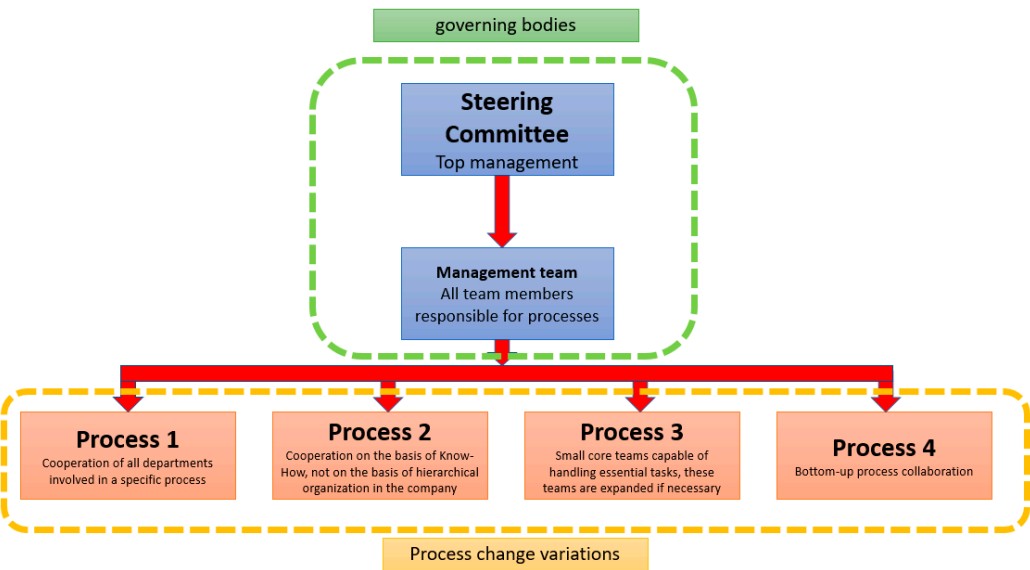

**Figure 8.** Organizational assurance of process change.

### 5.3. Monitoring of Key Indicators and Quantification of Changes

The objective of the third step is **the design of key performance indicators (KPI)** used to measure the overall long-term performance of the company. KPIs specifically help to determine a company's strategic, financial and operational achievements, especially when compared to the results of other businesses in the same sector. Key performance indicators measure a company's success against a set of goals, objectives, or industry competitors. KPIs can be financial, including net profit, gross profit margin, liquidity and cash availability, etc. The proposed KPIs are very clear to compile, and benchmarks are easily comparable with respect to the specifics of the company:

- Annual gross turnover in Euros (EUR);
- EBITDA in EUR;
- EBITDA in %;
- CAPEX;
- Cash Flow from Operations in EUR;
- Net debt in EUR;
- Net debt/EBITDA;
- Occupancy in the current year in %;
- Occupancy in the following year in %.

**Determining the planned target value for a given period**, which the company should try to meet in view of the development of the external environment, clearly presents how to manage the performance of the company in order to keep up with the overall goals. During the process of setting KPI goals, it is necessary to communicate frequently with your team to ensure strategic alignment of the entire business so that all team members are on the same page. The given specific values must be in close correlation with the short-term and long-term business plans of the company.

**KPI monitoring** is the definitive means of tracking the most important key performance indicators for increased business success. Enterprise resource planning (ERP) and Management information systems (MIS) provide organization and visualization of large data sets from databases. KPI monitoring requires automatic and online evaluation and presentation in the form of required reports, and dashboards at each management level of the company after fulfilling the conditions of Figure 9.

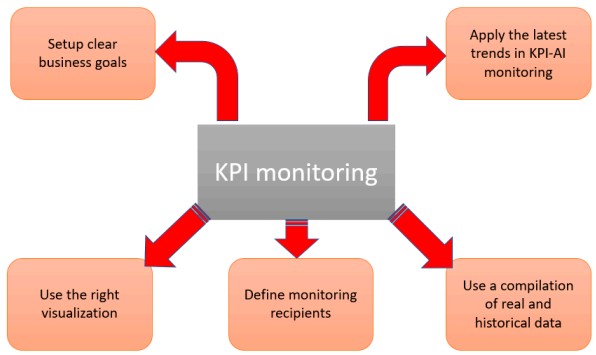

**Figure 9.** Graphic representation of the monitoring conditions of key indicators.

### 5.4. Monitoring Changes in the Company's Strategy

The goal of the fourth step is a process that identifies the consequences of a change in strategy that affects internal processes (Figure 10). The procedure for solving the process affected by the changes is described in detail in Section 5.2 (Methodology for changing internal processes) and graphically illustrated in Figure 6.

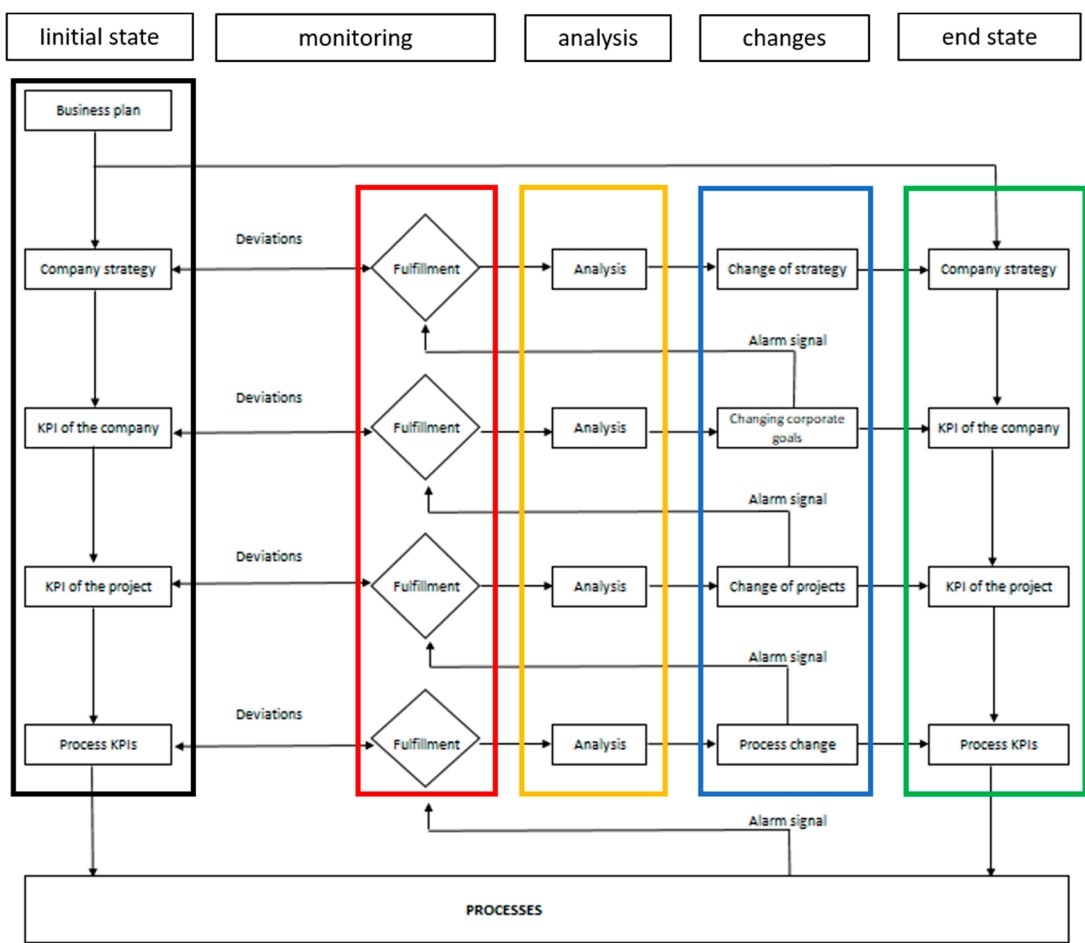

**Figure 10.** Graphical representation of the change monitoring design.

By measuring the performance of an organizational strategy, a business can decide whether to stay "on track" or adjust the strategic plan to adapt to changing market conditions. Monitoring and evaluation of the strategic plan provide an overview of possible failures and whether a change in direction in the overall vision of the company is necessary.

It is important for companies to regularly review their strategy and determine what is working and what is not.

When monitoring the impact of strategy changes, current and achieved values of key indicators are compared. It is evaluated whether the given key indicator is justified and whether it is important to monitor it. The impact of the strategy change is evaluated at four levels (Figure 10): company strategy; KPI of the company; KPI of the project; and process KPIs.

If there are changes in the strategic direction of the company, it is necessary to define new action plans. It is recommended to model the alternatives before introducing the change, which will be evaluated through a unified metric at the current level of knowledge. Monitoring is carried out at all the above-mentioned levels of management, i.e., j. strategy, enterprise, project, and process.

Monitoring any changes in management is carried out through indicators that are used to monitor pre-defined deviations in the required interval. If the set deviation is exceeded, the set processes are re-evaluated. The initialization of the change triggers a reassessment of the fulfilment of goals and KPIs at each level and is governed by the following rules:

- It is necessary to reassess the superior level whether the fulfilment of the goals will be sufficient;
- If there is no change in goals, the setting of the indicators will be left at the original value;
- If the current level or the superior level of the target is re-evaluated, it is necessary to recalculate the scope of the set boundaries of the indicators' targets at lower levels as well.

Changing a strategic goal is a process in which we change one or more strategic goals at the same time. This change can be only partial, or it can turn into a global change in the company's strategy. This process is triggered whenever at least one of the reasons for changing the strategy described in the previous chapter appears in the company or in its external environment. In the internal environment, the initiator of a change in the strategic goal is often non-fulfilment or significant deviations in the key indicators that were chosen for assessing performance at all levels of management.

Through research, we identified the causes of strategy changes and revealed some factors. In a global view, we can summarize the factors into groups:

- Changes in the political, economic, social, technological, ecological or legal environment that will affect the business and require certain adjustments that the business must make;
- Competitors have changed their approach or strategy, their meritorious actions can have an impact on whether it is viable for the company to continue with its own chosen strategy;
- New growth opportunities. For example, a firm may find that a closely related industry is growing rapidly and then decide to diversify the firm to take advantage of this new growth opportunity;
- Our own strategy is not working;
- Changes at the input level, such as input energy factors. Further, for example, relations with suppliers or customers. One-sided dependence on the supplier/customer base may begin to create greater bargaining pressure on the business, which may result in a change in business strategy to balance or eliminate the bargaining power of suppliers/customers.

*5.5. Quantifying the Impact of Changing Strategy and Internal Processes*

The goal of the fifth step is to evaluate the changes in strategy and the changes in internal processes caused by them by quantifying their impact on the company's key indicators. For this reason, it is ideal to establish the main key indicators of the business, which will not change over time. Of course, additional key indicators may be added, and these may change over time, but they will not have a great telling value about the

impact of changes. By correctly setting and naming the main key indicators, it is possible to monitor the quantification of the impact of strategy changes very clearly. For their better reporting ability and the creation of better prediction models, it is necessary to implement data collection and comparison on a regular basis. Key performance indicators suitable for comparison are: Annual gross turnover in EUR, EBITDA in EUR, BPT, Net debt/EBITDA.

### 5.6. Planning of Change

The last step—the implementation plan includes all the necessary steps that the implementation team should take in order to achieve the common goal. An implementation plan is a document that describes the necessary steps to achieve a common goal. The focus of any implementation plan is to ensure that the implementation team can answer the who? what? when? how? and why? questions before the actual implementation begins. Each action plan should combine strategy, processes and activities. A comprehensive action plan should include everything from the project strategy to the budget to the list of people working on the project. The action plan should include at least the following topics: description of the initiative, key results, team composition, assumptions, and risks.

## 6. Presentation of Results of the Systematic Procedure of Implementation of Strategy Changes

The proposed procedure of the methodology for the implementation of strategic changes into processes was continuously verified in the company Matador Automation, s.r.o. Next, we present the results of the steps of the verified procedure and the results achieved.

### 6.1. Implementation of Input Analyses

In the first phase of the procedure, the following analyses were performed. **Analysis of the input strategic visions** was carried out using rapid forecasts-brainstorming at the highest level of the strategic team. The result of this process is presented in Figure 11. It is a comprehensive overview of the strategic directions that the company should follow in the next five years.

**Figure 11.** Results of the analysis of input strategic visions.

Figure 12 shows the implemented analysis of the company's factors, which affect the negotiation position of the company in the industry using 5F Porter.

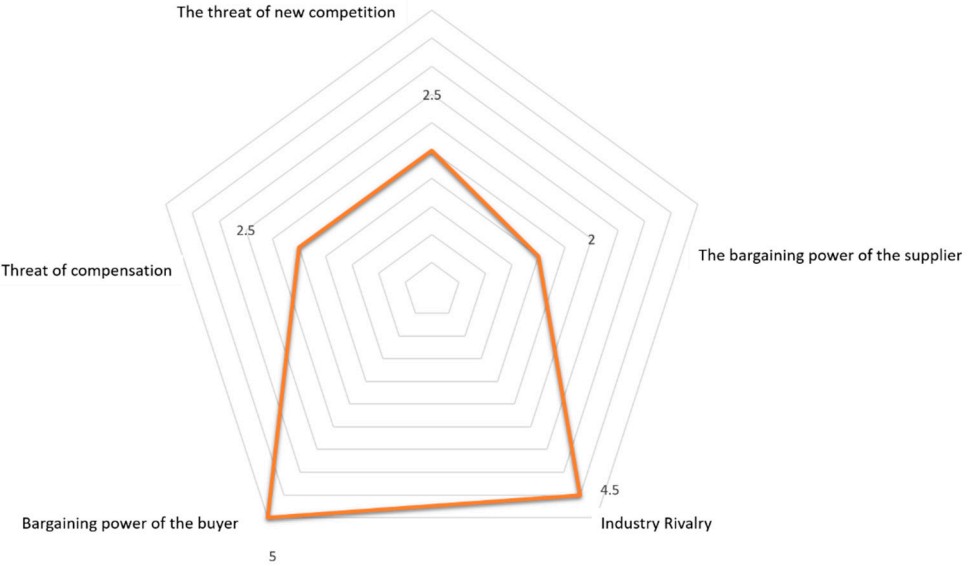

**Figure 12.** Results of the 5F Porter analysis.

**SWOT analysis** was used to analyze strengths and weaknesses, opportunities and threats (Figure 13).

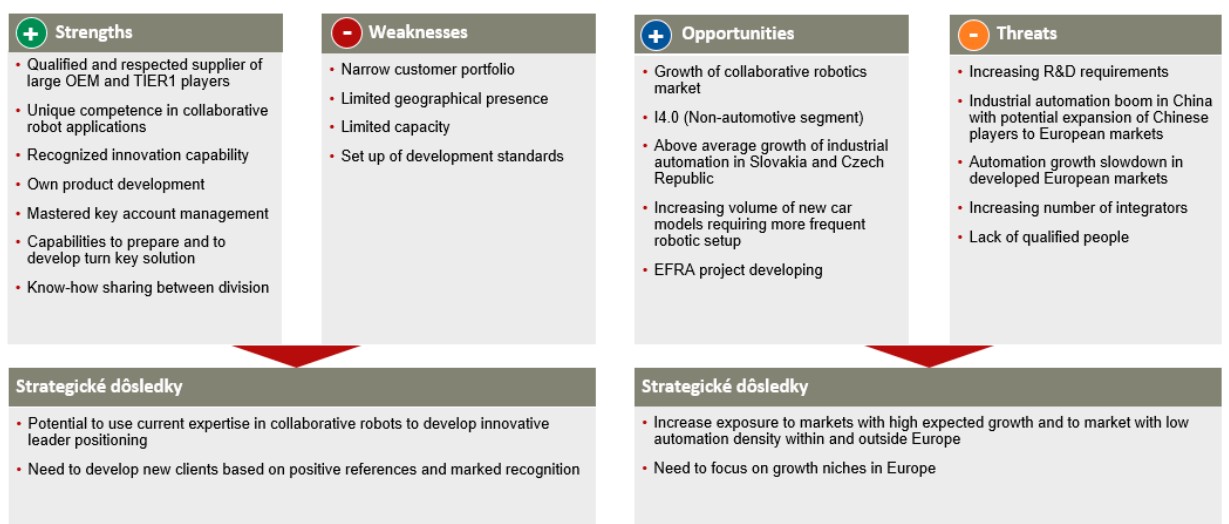

**Figure 13.** Results of the analysis of strengths and weaknesses.

**The initial analysis** of the competencies required for strategic growth in the recommended industries leads to the conclusion that the company has sufficient capabilities to implement the proposed strategic priorities successfully and organically (Figure 14).

| | Strategy move (What?) | Rationale (What?) | M&A (How?) | Organic (How?) | Rationale (How?) |
|---|---|---|---|---|---|
| **Expansion to laser welding** | • Complete current welding capabilities to catch up with core competitors | • Fast growing segment | | ✓ | • Adjacency, assumes hire of experienced experts |
| **Develop collaborative robotics** | • Develop collaborative robotics in order to establish a leading multi-niche competitive position | • Expected fast adoption from markets<br>• High-margin segment | | ✓ | • Further development of already partially mastered in-house capability |
| **Expansion to Germany** | • Expand to Germany with current and new business lines to serve the most advanced automotive OEMs | • European largest and most advanced market<br>• Geographic proximity | ✓ | ✓ | • Proven capability to penetrate new clients<br>• Potential to follow ESP and merge facilities |
| **Expansion to assembly** | • Develop assembly (and related handling) applications to cross-sell current accounts | • Sizeable market in automotive<br>• Expected fast growth | ✓ | ✓ | • Build option (cheaper) assumes hire of multiple experienced experts in assembly |

*(Priority order — top to bottom)*

**Figure 14.** Competence analysis.

**The customer satisfaction** survey was carried out in-house by the marketing department. The survey concerned five key customers of the company. The analysis shows that customers like to use the company's services and competencies. Negative assessment of the company's attitudes or performance was recorded in any of the survey areas. Customers also observed and appreciated the continuous improvement of the company in the last year, where the support of the company's professional training program, which focuses on HARD and SOFT skills for employees at different levels of management, was fully and positively manifested.

**The analysis of the internal environment** demonstrated sufficient internal capacity of the company for the necessary changes in processes due to changes in the company's strategy. The conclusions that emerged from this analysis can be summarized as follows:

- MATADOR Automation has its own complete set of IA skills;
- With >80% reported resource utilization, IA is utilizing its capacity;
- Acquisition of additional engineering and programming capabilities should be considered based on ongoing monitoring of the project pipeline;
- Capacities in accessory production and assembly and commissioning may remain outsourced even after further expansion of the business;
- The company has sufficient equipment and a wide infrastructure for the realization of strategic goals.

**The financial and economic analysis** demonstrated financial stability in the company and additional resources for the necessary implementation of strategy changes with an acceptable risk of temporary deterioration of the company's results. For the evaluation, verified ratio financial indicators were used: REVENUES—income, EBITDA—profit before interest, taxation, depreciation and amortization, PBT—profit before tax, NET DEBT—net debt, ROE—return on equity.

**The cost analysis of the projects** showed financial stability at the project level, but in 2017 it clearly showed the company's high dependence on external suppliers. This knowledge was taken into account in changes to the strategy and in the change in internal processes in the following years. The following project cost categories were verified for project monitoring (Table 2):

- Material complexity of projects;
- KOOP EXT—external cooperation;
- INT—internal performances;
- OPN—other direct costs (transport, travel, rent, etc.);
- SR—proper direction;

- Ratio of internal/external performances.

**Table 2.** Cost development in the proposed categories.

| Project | Act-17 | Increase | Act-18 | Increase | Act-19 | Increase | Act-20 | Increase | Act-21 |
|---|---|---|---|---|---|---|---|---|---|
| Material | 8.74 | | 11.51 | | 14.42 | | 11.87 | | 14.55 |
| KOOP EXT | 7.90 | | 5.20 | | 15.43 | | 12.20 | | 14.21 |
| INT | 2.38 | | 3.55 | | 2.77 | | 3.68 | | 4.13 |
| OPN | 0.11 | | 0.19 | | 0.84 | | 0.67 | | 0.73 |
| SR | 3.47 | | 3.52 | | 4.86 | | 5.27 | | 5.91 |
| Total | 22.61 | 1.10 | 23.97 | 1.60 | 38.32 | 0.88 | 33.69 | 1.17 | 39.52 |
| **Project** | **Act-17** | | **Act-18** | | **Act-19** | | **Act-20** | | **Act-21** |
| Material | 38.68% | | 48.03% | | 37.64% | | 35.23% | | 36.80% |
| KOOP EXT | 34.96% | | 21.69% | | 40.26% | | 36.21% | | 35.96% |
| INT | 10.55% | | 14.81% | | 7.23% | | 10.94% | | 10.45% |
| OPN | 0.47% | | 0.80% | | 2.18% | | 1.98% | | 1.84% |
| SR | 15.33% | | 14.67% | | 12.69% | | 15.63% | | 14.94% |
| Total | 100% | | 100% | | 100% | | 100% | | 100% |

### 6.2. KPI Monitoring and Quantification of Changes

Monitoring of the main key indicators was designed within the company, which provides:

- The evaluation of the achieved performances assuming the maintenance of the structure of indicators in the ERP system. It is implemented with the daily updating of data with connection to the ERP system automatically;
- Trend monitoring and comparison of currently achieved and target values of selected indicators.

The systems allow you to switch between individual time periods, which then allows you to follow the trend and compare the fulfilment of target values for selected processes. The green colour indicates fulfilment according to the defined level, and red indicates non-fulfilment of the required values of the planned indicators (Figure 15).

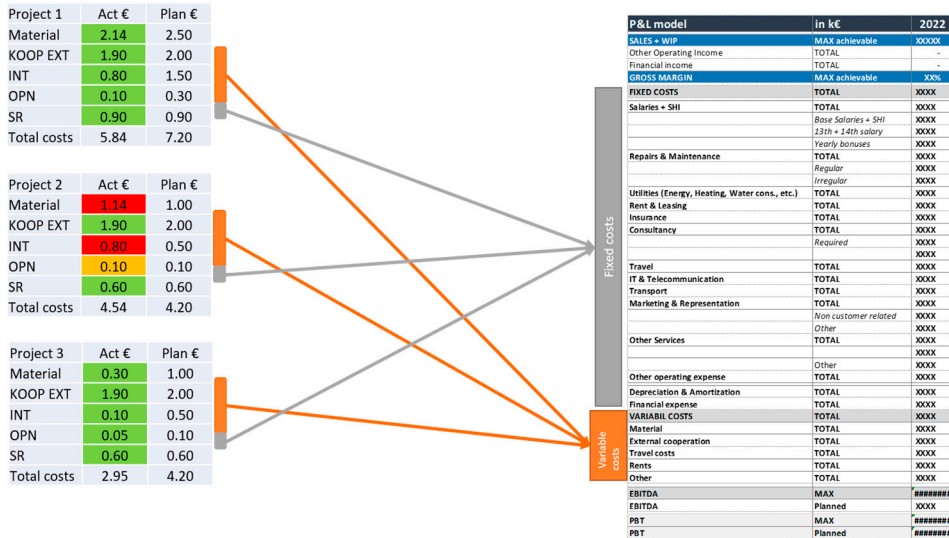

**Figure 15.** KPI monitoring.

This comparison shows the dependence of the company's KPI on the project's KPI, i.e., when any indicator on the project is exceeded, it is necessary to look at the relevant cost group from the level of the entire enterprise. It is necessary to carry out corrective measures on the project leading to the correction of this KPI, or to examine whether other projects

in that specific cost group of the company will create a sufficient reserve to cover this loss so that there is no deterioration of the company-wide KPIs and thus failure to fulfill the business plan in the relevant year.

The evaluation of indicators is carried out on a weekly basis at the level of top management. Documents and comments are prepared by the department of the economic director. Comments, explanations and corrective measures for KPI declines on projects are prepared by the project management departments of individual divisions, and are reported through the directors of specific divisions.

### 6.3. Monitoring Strategy Changes and Their Quantification

Monitoring changes in the company's strategy based on internal and external inputs, and regularly evaluating whether the company is on the right track or whether some strategic goal has lost its justification. It is a process that is carried out regularly (once a year) based on quantitative data obtained from the information system for the previous period. The output from monitoring changes in the company's strategy is a comprehensive report from the information system, where the individual strategic goals set for the relevant year in the company are evaluated based on the light principle (Figure 16). Meaning of colors:

- The green colour means that the strategic goal has been fulfilled to 90–100%;
- The yellow colour means that the strategic goal has been fulfilled to 50–90% and further work will be done to complete it;
- The red colour means that the strategic goal has lost its justification, and is cancelled as a goal.

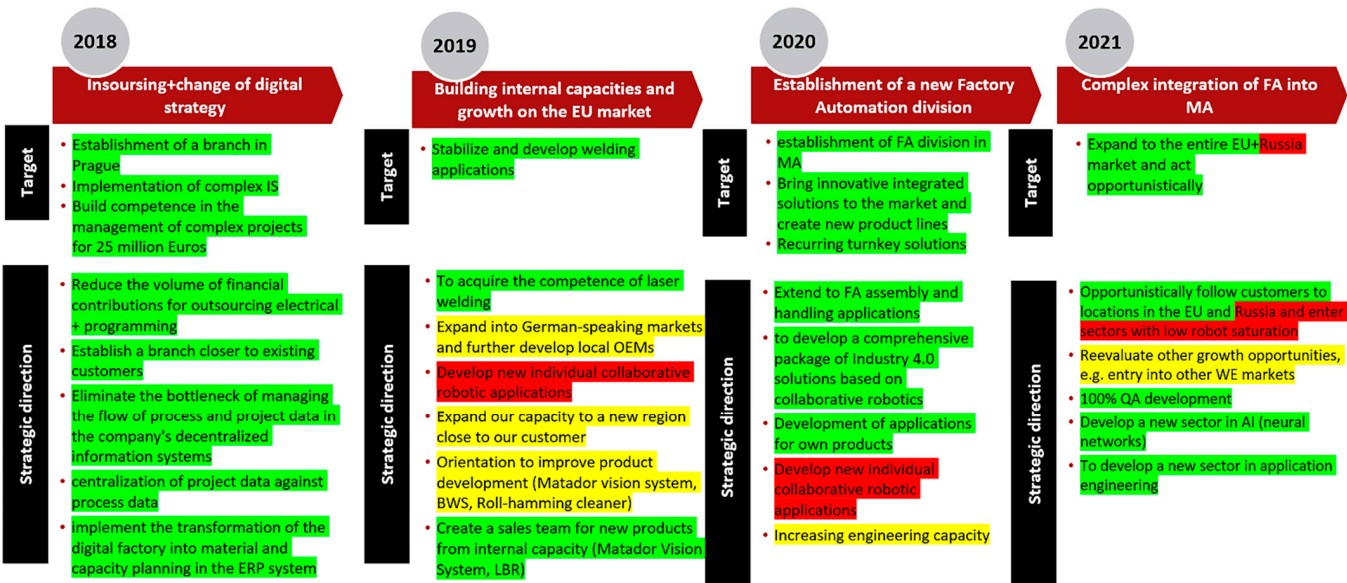

**Figure 16.** Strategy change monitoring report.

Inputs for new strategic goals were identified during the implementation of the study. In the internal environment they were caused by deviations with monitored KPIs. Inputs for new strategic goals from the external environment were identified as new market opportunities, or the impact of new technologies deployed in technical practice. Inputs for the cancellation of strategic goals came only from the external environment, and were caused by:

- A change in the market (low customer interest in collaborative robotics);
- Changing geopolitical conditions and sanctions (Russian Federation).

The proposed approach of tracking changes in the company's strategy made it possible to identify wrong directions in time and save resources that could be redirected to activities

that brought the desired financial effect, thus proving the correct setting of the strategy for the respective years. The impacts of the strategy changes are calculated in detail in Table 3 for the years 2017–2021.

**Table 3.** The impacts of the strategy changes for the years 2017–2021.

|  | Plan | Actuals | Plan | Actuals | Plan | Actuals | Plan | Actuals | Plan | Actuals |
|---|---|---|---|---|---|---|---|---|---|---|
|  | Act-17 | Act-17 | Act-18 | Act-18 | Act-19 | Act-19 | Act-20 | Act-20 | Act-21 | Act-21 |
| **Revenues (Sales + WIP) (MEUR)** | 24.0 | 23.0 | 23.3 | 25.3 | 26.0 | 40.5 | 35.3 | 35.8 | 39.6 | 42.0 |
| **EBITDA (MEUR)** | 1.8 | 1.9 | 2.1 | 2.1 | 2.1 | 3.1 | 3.2 | 3.9 | 4.3 | 3.8 |
| **EBITDA margin %** | 7.5% | 8.3% | 8.8% | 8.2% | 8.3% | 7.5% | 9.2% | 10.9% | 10.9% | 9.0% |
| **PBT (MEUR)** | 1.0 | 0.4 | 1.2 | 1.3 | 1.3 | 2.2 | 2.0 | 2.1 | 2.9 | 2.5 |
| **PBT % of Revenues** | 4.2% | 1.6% | 5.2% | 5.1% | 4.9% | 5.4% | 5.5% | 5.9% | 7.2% | 5.9% |
| **Net debt (MEUR)** | −8.0 | −9.6 | −1.1 | −1.1 | −4.0 | −1.1 | −3.5 | −4.3 | 3.3 | 5.2 |
| **Net debt/EBITDA** | −4.4 | −5.0 | −0.5 | −0.5 | −1.9 | v0.4 | −1.1 | −1.1 | 0.8 | 1.4 |
| **ROE** | 8.8% | 4.2% | 12.2% | 10.3% | 11.0% | 16.3% | 16.5% | 15.8% | 21.8% | 22.0% |

*6.4. Change in Internal Processes as a Result of Changes in Strategy*

As an example of a change in strategy, we present from the year 2018 the so-called Matador Automation's digital strategy. By changing the strategy of the company, it was necessary to resolve the change in the internal processes by the possibilities of the existing systems and the exact division of processes and responsibilities, which is enabled by the Product data management (PDM) system and the ERP system. Thanks to the modular architecture of the systems, each contributor to the process looks at the same data and only their view of it changes. The total integration of the Product Lifecycle Management (PLM) solution made it possible to track changes online, to have all data in precisely defined structures that copy reality, and to have access to data from anywhere in the world. Thanks to clearly defined business architecture, it is possible to track each item during its entire life cycle and all the necessary information. In addition to the standard main processes, supporting processes in the company such as Customer Relationship Management (CRM), Human resource management (HR) and the like were also integrated into the system. Support processes often interfere to a considerable extent with the main ones. As an example, we can cite HR management and the organizational structure, which is a direct source of data for capacity planning projects with a link to employee absences.

Thanks to the change in processes and the integration of systems it is possible to manage projects more efficiently, which increases efficiency while maintaining almost the same staffing. The systems provide a quick and easy view of projects for the needs of top management, but also a detailed breakdown of each item for the needs of employees responsible for modules and processes. After successful implementation in 2018, the company's turnover increased by 60% in 2019 and these results were achieved with only a 3% increase in human capacity. This clearly demonstrated what a huge contribution this fundamental strategic step had to internal processes. With essentially the same number of people thanks to the new digital strategy, the business was able to increase its sales by 60% and profit by 69%.

Another example of a change in strategy was in 2020, where the establishment of the new Factory Automation division was implemented. Before and after the establishment of the new division, it was necessary to resolve the changes in internal processes. At the end of 2019 and its subsequent full integration into the structures of Matador Automation in 2020, the quantitative effect of this strategic step and process resetting was fully manifested in 2021. As a result, it was able to saturate automation projects with higher added value. This new strategic step resulted in an increase in turnover by 3.7% but an increase in EBITDA of up to 22%.

*6.5. Action Plans*

Action plans are broad in scope and include all necessary steps. For individual strategy changes, changes in processes were proposed and divided into action plans with the objectives of changing internal processes: shortening the time needed to prepare a price offer; clear division of responsibilities during the process of preparing quotations; involvement of technical departments in the pricing process; a uniform form of price offers; a uniform form of exclusionary conditions; and implementation of the first CRM. Each strategy contains its own objectives of the given action plans (Table 4).

**Table 4.** Tracked goals of applied strategies.

| Tracked Objectives of the Digital Strategy Change: | Followed Objectives of PLM Implementation: |
| --- | --- |
| <ul><li>Elimination of process and project data flow management in decentralized information systems;</li><li>Ensuring sustainable growth of the company;</li><li>Paperless company;</li><li>Effective management of projects that multiply their turnover while maintaining almost the same staffing;</li><li>Online financial controlling;</li><li>Ability to track changes online;</li><li>All data in precisely defined structures that copy reality;</li><li>Access to data from anywhere in the world;</li><li>PDM + ERP linked online;</li><li>Just in time operations;</li><li>Integration of supporting processes in the company such as CRM, HR management, etc.</li></ul> | <ul><li>Exact data management and document management;</li><li>Real connection of the device on the line with the data in the system;</li><li>Accurate division of processes and responsibilities;</li><li>Business architecture;</li><li>Removal of duplicate solutions.</li></ul> |

## 7. Conclusions

New management initiatives require process management, and it is not possible to implement process management concepts according to old industrial age management models. Sustainable development is considered to be one of the responses to the destruction caused by industrialization, and it represents a major challenge for the field of business in all its aspects (population stabilization, food security, ecosystem resources, economy and industry).

The aim of this contribution is to present a study of the systematic procedure of implementing strategy changes into business processes for a project-oriented production organization. The study is based on a survey conducted in 100 Slovak companies, where the goal was to obtain a summary of factors that influence the success of internal process changes caused by changes in strategy. The experience gained from the business environment helped to define the requirements for the design of the methodology. After examining the current state and re-evaluating individual aspects, the authors proposed a procedure in six steps with the aim of maintaining the performance of the company's processes when changing strategy. The methodology has the task of ensuring the elimination of negative impacts by early identification of changes using KPI monitoring and timely intervention in processes through action plans so that currently ongoing projects are not threatened.

The described steps are designed to help consider the influence of all known external and internal factors. The methodology differs from previous procedures in that companies deal only with partial steps, and do not have an immediate view of the impact of business management during the implementation of a new strategy without a detailed investigation of the impact on internal processes. During the study, KPIs were selected and verified, which are easily comparable with competitive environments according to the benchmark.

We see the benefit of the study in the complex compilation of the methodology and its verification in a company that has the character of a project-based production organization, where the main goal is the successful implementation of projects. At the end of the contribution, there is a financial and economic presentation of the results of the application of the proposed systematic procedure for implementing strategy changes into processes. The approach of Matador Automation, s.r.o., was a big benefit, which influenced the implementation of the research and verification of the proposed methodology.

**Author Contributions:** Conceptualization, P.B. and J.C.; methodology, M.R.; software, J.C.; validation, K.S.; formal analysis, V.B.; investigation, J.C.; resources, J.C.; data curation, M.R.; writing—original draft preparation, J.C.; writing—review and editing, P.B.; visualization, J.C.; supervision, J.C. and K.S.; project administration, P.B.; funding acquisition, V.B. All authors have read and agreed to the published version of the manuscript.

**Funding:** This research study was supported by the VEGA 1/0524/22.

**Institutional Review Board Statement:** Not applicable.

**Informed Consent Statement:** Informed consent was obtained from all subjects involved in the study.

**Data Availability Statement:** The data presented are available on request from the corresponding author.

**Conflicts of Interest:** The authors declare no conflict of interest.

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
