# Peer review of "Impact of Strategy Change on Business Process Management"

_sustainability, doi:10.3390/su141711112_

Round 1

Reviewer 1 Report

Dear Authors,

Please consider the following suggestions for improvement of paper

1) The abstract should have more clarity about the objectives, methodology and outcomes of the study.

2) The introduction section should deliberate on the significance of strategic management for businesses, research problems and the need for the study.

3) The method section should be improved. Please refer to the articles from the journal. Also include the research methodology in detail with research problems, research designing, data collection, sampling etc.

4) The literature sections seem generalised. Please synthesise the literature and include the research gaps. Citations need to be improved. Also include the latest work from the domain

5) Analysis should provide detailed interpretations of the data.

6) Please also deliberate on the study's implications for businesses, government and society.

7) Please do not include the table in the conclusion section. Also, provide clear recommendations and limitations in the conclusion

8) Business Strategy is the broader aspect of the organisation. It is affected by the internal and external environment. Please try to incorporate the discussion section connecting it with your study

9) There are many bullet points Throughout the manuscript. Please try to minimise it by writing statements and explanations.

10) References are older and very limited. Please revise

Reviewer 2 Report

Overall a very interesting topic; while the topic is indeed of great value to both academia and the practitioner communities, there remain a few issues that ought to be addressed before this paper can be accepted.

I have only a few concerns about the paper and some suggestions that maybe the authors could consider:

1.     In the 'Introduction' section, the proposed research gap and the stated objectives do not meet the criteria of proper synergy. Please make the research gap and the research objectives consistent with each other.

2.     I think in the “Literature” section, you can add paragraph that you can brief the structure of article (see https://doi.org/10.3389/fnins.2020.594566).

3.     I think that in the “Introduction” section, you can add some sentences and add some type of technological innovation tools, which can use effectively in business strategy and management to improve your section. I suggest some references which can be beneficial for this, as follows:(https://doi.org/10.1155/2019/1976847; https://doi.org/10.3991/ijoe.v18i08.31959; https://doi.org/10.3389/fnins.2020.594566).

4.     The authors should explicitly state the novel contribution of this work and its similarities and differences with their previous publications.

5.     The authors need to clearly articulate the academic and practical implications of this study. I would suggest writing a paragraph in the Discussion and conclusions section for the implications. Also, state a few of the key implications at the end of the 'Introduction' section.

6.     For readers to quickly catch your contribution, it would be better to highlight major difficulties and challenges and your original achievements to overcome them in a clearer way in the abstract and introduction.

7.     How could/should futures studies improve the model?

If these revisions can be made in the manuscript, I believe that this study can be accepted for publication.

I wish the authors all the very best with this study.

Author Response

Overall a very interesting topic; while the topic is indeed of great value to both academia and the practitioner communities, there remain a few issues that ought to be addressed before this paper can be accepted.

EXPLANATION: Dear reviewer, Thank you very much. We are very grateful for your kind and polite appreciation of our work. We have highlighted the changes in green.

I have only a few concerns about the paper and some suggestions that maybe the authors could consider:

  1. In the 'Introduction' section, the proposed research gap and the stated objectives do not meet the criteria of proper synergy. Please make the research gap and the research objectives consistent with each other.

EXPLANATION: Thank you for all your valuable comments. We tried to incorporate all comments into the manuscript. Thanks for the guidance to improve the manuscript. We tried to refine the introduction, where we added the aims, methodology and results of the study.

  1. I think in the “Literature” section, you can add paragraph that you can brief the structure of article (see https://doi.org/10.3389/fnins.2020.594566).

EXPLANATION: Thank you for all your valuable comments. We tried to use this structure.

  1. I think that in the “Introduction” section, you can add some sentences and add some type of technological innovation tools, which can use effectively in business strategy and management to improve your section. I suggest some references which can be beneficial for this, as follows:(https://doi.org/10.1155/2019/1976847; https://doi.org/10.3991/ijoe.v18i08.31959; https://doi.org/10.3389/fnins.2020.594566).

EXPLANATION: Thank you for all your valuable comments. We added this publication as you advised us.

  1. The authors should explicitly state the novel contribution of this work and its similarities and differences with their previous publications.

EXPLANATION: : Thank you for all your valuable comments. In the conclusion, the authors stated a new contribution of this work.

  1. The authors need to clearly articulate the academic and practical implications of this study. I would suggest writing a paragraph in the Discussion and conclusions section for the implications. Also, state a few of the key implications at the end of the 'Introduction' section.

EXPLANATION: Thank you for all your valuable comments. The authors restated the implications of this study in the discussion and conclusions section. At the end of the “Introduction” section, we stated several key implications.

  1. For readers to quickly catch your contribution, it would be better to highlight major difficulties and challenges and your original achievements to overcome them in a clearer way in the abstract and introduction.

EXPLANATION: Thank you for all your valuable comments. In the abstract and introduction was changed.

  1. How could/should futures studies improve the model?

EXPLANATION: Thank you for all your valuable comments. How could/should futures studies improve the model? Future studies could also focus on other characteristics of manufacturing enterprises. Future study concerns project-oriented manufacturing.

If these revisions can be made in the manuscript, I believe that this study can be accepted for publication.

I wish the authors all the very best with this study.

Dear reviewer, thank you for your recommendation and kind suggestions. All the best to you.

Best regards, authors.

Reviewer 3 Report

Dear authors, 

Thank you for providing the chance to review your article. The theme is interesting and useful. The contribution is in presenting a proposal for a systematic procedure for implementing strategy changes into business processes for a project-oriented production and the authors present how this proposal is viable on a case study example. However, I believe that in general this paper does not represent a true scientific research paper-article. It is written more like a book chapter or overview paper. The authors provide many theoretical concepts, definitions, listings – many without proper references. It is not really clear what is the contribution of this paper. The authors did not provide and identified any research gap, did not provide insight into what we now, what we do not know and how their paper contributes to reduction of this gap. The contribution of the paper is vague and unclear. The writing style is confusing as there are many too long sentences throughout the paper.

The methodology is not clear, especially the one related to your case study data. The question is also about the data provided for the case study -  how was this data collected, do you have a permission to list many detailed information about the company studied? The conclusion and discussion of the paper are quite weak. The paper misses insight into practical and theoretical contribution of the paper. The reference list is not adequate.

Author Response

EXPLANATION: Dear reviewer,

thank you for your recommendation and kind suggestions We have highlighted the changes in green. Thanks for the guidance to improve the post. We tried to refine the research design. We have tried to edit parts of the article. We believe that it now makes sense and is beneficial to businesses on the way to change processes due to a change in strategy. We have permission to publish data from the director of the company. We see the filling of the gap in research in the establishment and verification of steps that define KPIs, monitor their change with the designed monitoring and warn the management in time on how they react to a change in strategy. The goal is to ensure that the effectiveness of ongoing projects is not disrupted and that the effects of ongoing changes are known.

All the best to you.

Best regards, authors.

Round 2

Reviewer 1 Report

Authors have incorporated the suggested changes.

Author Response

Dear reviewer,

we are very grateful for your kind and polite appreciation of our work. We tried t improve our paper

Reviewer´s comment 1: Authors have incorporated the suggested changes.

Author´s answer: Dear reviewer, thank you very much for thinking that we have incorporated the suggested changes. We have added information to clarify the purpose of the research and its results. We have made other revisions which are marked in green colour.

Yours sincerely,

authors

Reviewer 3 Report

Thank you for the revised version of your paper. Still, the paper continues to miss a more clear insight into your methodology and better explanation of your selection criteria, research process and similar (for instance you mention questionnaire of companies in Slovakia - was there more than one company - than how is his a one company case study? Your discussion and conclusion continue to be weak and need to be upgraded as previously suggested. Overall, the paper still has a sense of being to vague and general.

Author Response

Dear reviewer,

we are very grateful for your kind and polite appreciation of our work.

Reviewer´s comment 1: Thank you for the revised version of your paper. Still, the paper continues to miss a more clear insight into your methodology and a better explanation of your selection criteria, research process and similar (for instance you mention a questionnaire of companies in Slovakia - was there more than one company - than how is his a one company case study? Your discussion and conclusion continue to be weak and need to be upgraded as previously suggested. Overall, the paper still has a sense of being to vague and general.

Author´s answer: Dear reviewer, thank you very much for thinking that we have incorporated some of the suggested changes. We are very grateful for your kind and polite appreciation of our work. We tried to improve our paper. Based on your requests, we have added information to clarify the purpose of the research and its results. We have made other revisions which are marked in green colour.

Yours sincerely

authors

Round 3

Reviewer 3 Report

Thank you for the revised version.